# Long-Term Corrosion Testing of Zy-4 in a LiOH Solution under High Pressure and Temperature Conditions

**DOI:** 10.3390/ma14164586

**Published:** 2021-08-15

**Authors:** Diana Diniasi, Florentina Golgovici, Alexandru Horia Marin, Aurelian Denis Negrea, Manuela Fulger, Ioana Demetrescu

**Affiliations:** 1Institute for Nuclear Research Pitesti, POB 78, Campului Street, No. 1, 115400 Mioveni, Romania; diana.diniasi@nuclear.ro (D.D.); marin.alexandru.horia@gmail.com (A.H.M.); manuela.fulger@nuclear.ro (M.F.); 2Department of General Chemistry, Faculty of Applied Chemistry and Materials Science, University Politehnica of Bucharest, Splaiul Independentei Street, No. 313, 060042 Bucharest, Romania; ioana.demetrescu@upb.ro; 3Regional Center of Research & Development for Materials, Processes and Innovative Products Dedicated to the Automotive Industry (CRC&D-AUTO), University of Pitesti, Doaga Street, No. 11, 110040 Pitesti, Romania; denis.negrea@upit.ro; 4Academy of Romanian Scientists, 3 Ilfov, 050094 Bucharest, Romania

**Keywords:** corrosion/oxidation, zircaloy-4, XPS, EIS

## Abstract

The fuel cladding is one of the most important structural components for maintaining the integrity of a fuel channel and for safely exploitation of a nuclear power plant. The corrosion behavior of a fuel cladding material, Zy-4, under high pressure and temperatures conditions, was analyzed in a static isothermal autoclave under simulated primary water conditions—a LiOH solution at 310 °C and 10 MPa for up to 3024 h. After this, the oxides grown on the Zy-4 sample surface were characterized using electrochemical measurements, gravimetric analysis, metallographic analysis, SEM and XPS. The maximum oxide thicknesses evaluated by gravimetric and SEM measurements were in good agreement; both values were around 1.2 µm. The optical light microscopy (OLM) investigations identified the presence of small hydrides uniformly distributed horizontally across the alloy. EIS impedance spectra showed an increase in the oxide impedance for the samples oxidized for a long time. EIS plots has the best fit with an equivalent circuit which illustrated an oxide model that has two oxide layers: an inner oxide layer and outer layer. The EIS results showed that the inner layer was a barrier layer, and the outer layer was a porous layer. Potentiodynamic polarization results demonstrated superior corrosion resistance of the samples tested for longer periods of time. By XPS measurements we identified all five oxidation states of zirconium: Zr^0^ located at 178.5 eV; Zr^4+^ at 182.8 eV; and the three suboxides, Zr^+^, Zr^2+^ and Zr^3+^ at 179.7, 180.8 and 181.8 eV, respectively. The determination of Vickers microhardness completed the investigation.

## 1. Introduction

For materials used in the sustainable energy domain, investigations of properties are continuous, to enhance selection [1]. In the nuclear field, the corrosion processes of all metallic materials have determining roles in efficiently and safely operating a nuclear power plant (NPP), and their oxidation processes are largely investigated as functions of the environment for their industrial exploitation [2,3,4,5]. Zirconium-based alloys serve the nuclear industry as metallic tubes called cladding, where the nuclear fuel element is located and assembled. They are exposed to an aggressive mediums of water at high pressures and high temperatures (255–310 °C, 10 MPa, 1 wt. ppm Li), in addition to high fluxes of energetic neutrons. In this environment, the mechanical strength and corrosion behavior of the materials are of extreme importance. The candidate materials that meet such criteria, and have high thermal conductivity and low absorption cross-sections for thermal neutrons, are Zr alloys [6]. For low neutron absorption, the alloys must have no hafnium [7]. Being used as fuel cladding materials in pressurized heavy water reactors (PHWR), Zr alloys represent the first containment barrier to fission products. Their mechanical integrity has to be ensured during their life cycles.

The Zircaloys [8] Zircaloy-2 and Zircaloy-4 have appropriate chemical compositions. In comparison to Zircaloy-2, Zircaloy-4, which was introduced later, is a Ni free material and has better performance related chiefly to its lower hydrogen absorption under water corrosion [9]. As was presented recently in the literature, Zircaloy-4’s parameters for fuel rod cladding [6] indicated a doubling of thermal conductivity for a temperature increase from 1000 to 4000 C. In the same conditions, the Young’s Modulus change was only from 88 to 72 GPa. Zirlo is another alloy of Zr with less Sn and Ni, and no Cr. It has an amount of Nb close to 1% [10].

It is well known that for Zr alloys and other engineering materials, the corrosion resistance and mechanical strength are closely related to the alloying elements and their amounts [10,11,12]. Sn promotes solid solution strengthening in relatively small amounts, but excessive content is a factor determining an increase of the corrosion rate [13]. The elements Fe, Cr and Nb are always precipitated from the α-Zr matrix, forming Zr (Fe,Cr)_2_ and β-Nb phases, able to improve the mechanical properties of alloys at the expense of the corrosion resistance [14,15]. As expected, the low-Sn Zr alloys are good candidates for enhancing the corrosion resistance of Zr cladding alloys in nuclear reactors, presenting both high corrosion resistance and high strength. In this context, as a strategy for improving cladding materials, a new alloy series of Zr-0.25Sn-0.36Fe-0.11Cr-xNb (x = 0.4~1.2 wt.%) was developed and investigated [16]. Having various amounts of Nb, such alloys are in fact modified from Zirlo alloys and exhibit a single hexagonal α-Zr structure [17]. Recent studies showed that particles precipitated in the α-Zr matrix have a uniform distribution, and the total amount of precipitates increases with the Nb content from x = 0.4 to x = 1.2. The particle sizes in all these alloys are almost constant, being about 50–100 nm. In the Z-0.4Nb alloy, the precipitates are scarce, indicating that the alloying elements are almost soluble in an α-Zr matrix.

In these conditions, the corrosion kinetics of Zircaloy-4 shows a drastic acceleration phase referred as “high burnup acceleration,” which occurs for fuel burnups above 35 GWd/tU [18].

Density functional theory (DFT) and autoclave corrosion experiments in 360 °C water were performed to investigate the influences of several elements, including Sb, Sc, Nb and Sn, on the corrosion and hydrogen pick-up of Zr-alloys [19].

Therefore, the corrosion of Zircaloy-4 is one of the main limiting factors of the fuel rod’s lifetime. Several authors suggested some theories for the kinetic acceleration at high burnup as the evolution of the tin distribution in the alloy in service [20,21], or the irradiation effect of the material leading to the amorphization of the Zr (Fe, Cr)2 precipitates and the iron dissolution in the matrix [22,23]. Other authors say that the precipitation of numerous hydrides at the metal–oxide interface in the metallic part of the cladding is the basis of kinetic acceleration [24,25].

The actual trend regarding the enhancement of nuclear plants’ performances (higher coolant temperatures, higher burnups and expanding NPP lifetimes) puts severe strains on the materials in the reactor. Two of the main limiting factors for the lifetime of the fuel cladding are the oxidation and the hydrogen pick-up, which are consequences of the water-side corrosion process in the reactor.

The objective of this work is to contribute to knowledge about future materials for nuclear power plants—more specifically [26], to describe the corrosion behavior of a fuel cladding material, Zircaloy-4, under high pressure and temperatures conditions.

## 2. Materials and Methods

### 2.1. Materials

Zircaloy-4 (Zy-4) tube alloy is studied in this paper. The chemical composition of this alloy was established from EDX analysis performed with an energy dispersive spectra detector (EDS), attached to a Hitachi SU 8230 scanning electron microscope (Hitachi, Tokyo, Japan) and is presented in Table 1. The Zircaloy-4 tube with an outer diameter of 13 mm and a wall thickness of 0.45 mm was cut into pieces of 20 mm length and then half-cut. No specific surface preparation was performed.

Autoclave corrosion was accomplished in a 1 L static isothermal autoclave under simulated primary water conditions—a 0.585 M LiOH solution at 310°C and 10 MPa for up to 3024 h. LiOH was added to the cooling water to provide an alkaline pH for inhibiting the deposition of corrosion products on the fuel cladding and for reducing the corrosion rates of structural materials.

The alloying elements in the Zr alloy for the nuclear reactor can affect its properties It is known that tin can improve the strength of Zr, reduce its plasticity and improve its creep and corrosion [6]. The LiOH solution had at 25 °C a conductivity of 71 µS cm^−1^, and its dissolved oxygen content was maintained below 2 ppm by thermal degassing at 100 °C. The pH of testing solution was about 10.5 (measured at room temperature, about 25 °C). Samples were systematically removed (at every 21 days) from the autoclave for weight gain measurements and occasionally archived for microstructure examination and electrochemical testing. After cleaning with acetone and drying, the weight gain of each sample was measured. The autoclaved solution was replaced with a fresh solution after each inspection. Weight gain due to oxidation was measured using a balance, providing a precision of ±10^−4^ mg.

### 2.2. Morphological and Structural Surface Analysis

After exposure to the LiOH solution, the oxides formed on the Zy-4 sample surface were characterized using gravimetric analysis, metallographic analysis, SEM and XPS. Metallographic analysis was performed using the Olympus BX51M optical microscope (Olympus Corporation, Tokyo, Japan) to highlight Vickers microhardness. SEM/EDS investigations were carried out using a HITACHI SU5000 field emission scanning electron microscope (Hitachi, Tokyo, Japan) equipped with an energy-dispersive X-ray analyzer (EDS, Oxford Instruments, Oxford, UK) operated at an accelerating voltage of 30 kV. To determine the thickness of the oxide layer, small pieces were cut from the samples and wrapped in copper foil, embedded in conductive cupric resin and ground (P #4000 μm). The hydrides were highlighted by chemical etching in a solution composed of 45 mL HNO_3_ (67%), 45 mL H_2_O_2_ (30%) and 7 mL HF (30%) for 20 s. The Vickers microhardness (MHV_0_._1_) was determined with an OPL tester in an automatic cycle.

XPS measurements were performed on autoclaved Zy-4 alloys with an Escalab 250 system (Thermo Scientific, East Grinstead, UK) equipped with a monochromated Al Kα (1486.6 eV) X-ray source and a base pressure in the analysis chamber of 10^−8^ Pa. The acquired spectra were calibrated with respect to the C1s line of surface adventitious carbon at 284.8 eV. An electron flood gun was used to compensate for the charging effect in insulating samples.

### 2.3. Electrochemical Tests

Electrochemical measurements were performed using a PARSTAT 2273 computer-controlled electrochemical measurement system (Princeton Applied Research, AMETEK, OakRidge, TN, USA) with a conventional three-electrode electrochemical cell consisting of a working electrode (Zy-4 sample), a saturated calomel reference electrode (SCE) and two auxiliary electrodes (graphite rods). The electrochemical tests were carried out at room temperature (22 ± 2 °C). Electrochemical impedance spectroscopy measurements were performed in a chemically inert solution with pH = 7.26 (0.05 M boric acid with 0.001 M borax solution), which did not affect the oxide layer features. For electrochemical impedance spectroscopy tests and for open circuit potential measurements, a LiOH solution with a pH of 10.5 was used as the electrolyte.

The electrochemical parameters, characterization and determination of the oxide layer behavior were realized by potentiodynamic measurements, electrochemical impedance spectroscopy and open circuit potential variation.

Potentiodynamic measurements were made at room temperature at a scan rate of 0.5 mV·s^−1^ and a range from −250 to 1000 mV relative to the open current potential (OCP), in a specific primary circuit solution (LiOH solution, pH 10.5). The behavior of the material with generalized corrosion was observed by open circuit potential measurements in the same electrolyte.

Electrochemical impedance spectroscopy (EIS) tests were performed at open circuit potential (OCP) with an amplitude of 10 mV in the frequency range from 100 mHz to 100 kHz after OCP stabilization. To obtain quantitative data, the experimental EIS results were simulated with equivalent electrical circuits as appropriate models using ZView 2.90c software (Scribner Associates Inc., Southern Pines, NC, USA).

## 3. Results and Discussion

### 3.1. Oxidation Kinetics

Weight gain measurements were performed from 0 to 3024 h, for the oxidation experiment carried out at 310 °C and 10 MPa. In Figure 1, the weight gain data as a function of exposure time for Zy-4 samples in LiOH solution are represented.

As can be seen from Figure 1, all samples gained weight during oxidation. The oxidation process was seen to follow a rate law described by:(1)$$\Delta W=k_{p}*t^{n}$$
where ΔW is the oxide weight gain (mg/dm^2^), k_p_ is the rate constant, t is exposure time (h) and *n* is the exponent. If *n* is close to 0.3, the law is cubic; if *n* is close to 0.5, the law is parabolic; and if *n* is close to 1, the law is linear [27]. After a few microns of oxide growth, an increase of the corrosion rate took place as the protective layer broke down. After this transition process, a post-transition with a reduced corrosion rate was again observed. In fact, this process repeated in a cycle [28].

The oxidation constants were determined by fitting the data with Equation (1). The obtained k_p_ value and R-squared value of trend (R^2^) of the plot are also summarized in Table 2. We see that *n* is closer to 0.3, so the corrosion law is cubic. The R-squared value was calculated to determine the reliability of the trend. By analyzing weight gain data, following the fitting of the curve, it was established that, in the case of Zy-4 alloy, the oxidation process is best described by cubic kinetics (*n* = 0.199).

Based on the values obtained from the weighing of the samples, the oxide thickness was calculated at different periods of oxidation. Oxide film thickness was calculated from the weight gain according to Equation (2) [29].
(2)$$T=\frac{W_{2}-W_{1}}{A\times M_{O_{2}}}\times M_{ZrO_{2\;}}÷D_{ZrO_{2}}$$
where *T* is oxide film thickness, *W*_1_ and *W*_2_ are weight before and after oxidation, A is the surface area of the sample, $M_{O_{2}}$ is the mass number of O_2_, $M_{ZrO_{2\;}}$is the mass number of $M_{ZrO_{2\;}}$ and $D_{ZrO_{2}}$ is ZrO_2_ density.

Figure 2 shows the oxide thickness behavior of the Zy-4 samples in the LiOH solution at 310 °C and 10 MPa.

The chemistry of Zircaloy-4 metal–oxide interface data was compared to other literature results [30].

From Figure 2, we can see that the oxide thickness increased with the oxidation time. Typically, Zy-4 oxidation (in high-temperature water, steam or air) occurs in two stages: pre-transition and post-transition. In the first stage, the kinetics of the oxidation process follow a cubic law, predicted by the Wagner/Hauffe theory [31]. The appearance of the oxide is blackish at this stage, and grains are small and equiaxed.

It was observed that at some critical thickness (2–3 µm), which is a function of temperature and other variables, transition to linear growth with time occurs. Thus, the oxide becomes less protective [32,33].

The pre-transition oxide layer formed on the surface of a zirconium alloy has distinct compositions from the outside inwards. The outer layer is predominantly columnar monoclinic ZrO2. The second region is a 50–80 nm layer with equiaxed tetragonal ZrO_2_, and the third is a 100–200 nm layer of sub-stoichiometric Zr-oxide. The final region is α-Zr metal.

### 3.2. Morphological and Structural Characterization

#### 3.2.1. Metallographic Analysis (Optical Microscopy)

There is a common understanding that only atomic hydrogen that is generated by cladding corrosion is picked-up by Zy alloys and not the dissolved molecular hydrogen that is added into the reactor coolant [34]. When the solubility limit for picked-up hydrogen (about 100 mg/kg at 330 °C) is exceeded in the Zr alloys, the excess hydrogen precipitates as Zr-hydride. Precipitated hydrides in Zr alloys are predominantly in the delta phase (ZrH_x_, x ~ 1.66) [35,36]. The formation of elongated zirconium hydride platelets during corrosion of nuclear fuel cladding is linked to its premature failure due to embrittlement and delayed hydride cracking. Figure 3 shows representative hydrides in Zy-4 microstructure.

The optical light microscopy (OLM) investigations identified the presence of small hydrides uniformly distributed horizontally across the alloy. This resulted in a very fine microstructure. In Figure 3, we can observe growth in the density of hydrides from 504 testing hours to 1512 and 3024 testing hours. However, the hydride precipitation process was not accelerated from 1512 to 3024 testing hours. The inhibition of hydride precipitation can be explained by the ZrO_2_ barrier layer at the oxide–metal interface [37].

From the microhardness tests, which were from oxide to alloy, it could be observed there was a slight decrease of Vickers hardness values, from 226 ± 3 kgf/mm^2^ for unoxidized samples to 223 ± 2 kgf/mm^2^ for the samples oxidized for 504 h. Then, the values were 214 ± 2 kgf/mm^2^ for the samples oxidized for 1512 h, and finally, 211 ± 2 kgf/mm^2^ was the hardness for the samples oxidized for 3024 h.

#### 3.2.2. Scanning Electron Microscopy (SEM) Measurements

In order to obtain information about the morphologies of the thermally oxidized and unoxidized Zy-4 samples, SEM microscopy was applied. Five measurements were performed each time. Figure 4 presents the morphologies of unoxidized and three oxidized zircalloy surfaces.

The surface morphologies of the alloys before and after their interactions with the LiOH solution can be observed in Figure 4. The autoclaving process combined with the LiOH exposure induced significant changes in surface area compared to the unheated sample. The hydrides were dispersed relatively homogeneously across the surfaces. By varying the autoclaving time, strip-like hydrides occurred for all samples subjected to the LiOH environment. This was more pronounced after longer periods of exposure (Figure 4b–d). Moreover, the 3024 h autoclaving time generated hydride cracking, which is visible in Figure 4d.

By performing SEM on cross-sections of the samples, the thickness of the oxide layer present on the surface of each sample could be measured. The measurements can be seen in Figure 5.

As for the unoxidized sample, from the EDS analysis we obtained the compositions of the surfaces of the three oxidized samples for three different periods of time (Table 3). The oxygen peak appeared in the all spectra, but peaks for Zr, Sn, Cr and Fe also appeared due to the composition of the alloy.

The results from the quantitative analysis indicate increasing oxygen content with increasing oxidation time (Table 3, Figure 6). Thus, when exposing samples to our LiOH solution for 504 and 3024 h, the oxygen content increased from 25% to 28%. This was due to an oxide layer forming on each surface and increasing in thickness.

In order to better observe the element distributions on the Zy-4 surfaces, mapping of the elements was performed. A uniform distribution of oxygen can be observed from Figure 7.

The presence of an oxide layer on the surface of each of the three thermally oxidized samples can be observed in Figure 8, which shows the EDS compositional profiles of cross-sections through the oxide layers grown on the Zy-4 samples.

#### 3.2.3. XPS Measurements

XPS investigations have been carried out to assess the bonding states of atoms on the samples’ surfaces, and after, quantitative analysis to find the relative concentrations of elements and chemical states. Narrow-range XPS spectra of the most prominent photoelectron peaks were collected in order to establish the chemical modifications which may occur during the autoclaving of zirconium alloys at 310 °C and 100 bar pressure for 504 or 3024 h.

An overview of zirconium chemistry as a function of autoclaving conditions is portrayed in Figure 9, revealing different chemical behavior resulting from different lengths of time.

In order to establish the chemical states of zirconium, the spectra were deconvoluted into component peaks using curve-fitting procedures, as demonstrated in Figure 10a,b.

The Zr3d band-like spectrum of the autoclaved Zy-4 alloy accommodates a mixture of chemical states, and hence, by spectral deconvolution, the chemical species associated with zirconium were separated and identified, illustrating all five oxidation states of zirconium (Figure 10a, Table 4). Thus, Zr0 is located at 178.5 eV, and fully oxidized zirconium in its Zr^4+^ was found at 182.8 eV [38,39,40,41]. The three suboxides, Zr^+^, Zr^2+^ and Zr^3+^, peak at 179.7, 180.8 and 181.8 eV, respectively [38,39,41].

After increasing the heat-treatment duration from 504 to 3024 h (Figure 10b), which translates into six times the autoclaving period, the chemical behavior of zirconium differed quantitatively (Table 4). There was a greater Zr^3+^ contribution (Zr_2_O_3_), from 31.4% to 54.5%, accompanied by a decrease of metallic zirconium, from 9.3% to 3.0%. At the same time, Zr^4+^ (ZrO_2_) vanished (Figure 10b, Table 4). The oxygen concentration on the surface remained constant though (Table 5).

Normally, with increasing oxidation time, from 504 to 3024 h, one would expected a chemical tendency towards full oxidation of zirconium, possibly indicated by the appearance of a large ZrO_2_ contribution. However, in this case, zirconium’s behavior was different, as it showed complete disappearance of the ZrO_2_ component (Figure 10a, Table 4).

On the other hand, other studies confirm the formation of suboxides of zirconium on the surface, along with ZrO_2_, during zirconium oxidation by water vapor or an oxygen atmosphere at room temperature and normal pressure [38,39]. A different approach was proposed by Kaufmann [42], who studied the oxidation mechanism of Zy-4 alloy at temperatures between room temperature and 500 °C, and reported a reduction of ZrO_2_ to Zr metal, which occurred at temperatures above 300 °C, followed by oxygen diffusion into the bulk. Moreover, West [43] observed the dissolution of ZrO_2_ into the bulk when 400 °C was used. ZrO was initially obtained by exposing pure zirconium samples to oxygen at 310 °C.

The interaction of oxygen with zirconium surface at room temperature was also considered by Morant [41], who suggested the occurrence of a ZrO_2_ phase accompanied by suboxides with increasing oxygen content and the conversion of adsorbed oxygen into lattice oxygen, revealing the presence of oxidation states Zr^+^, Zr^2+^, Zr^3+^ and Zr^4+^. Similarly, the oxidation of zirconium with oxygen under room temperature conditions showed the formation of intermediate oxide states of zirconium atoms in addition to ZrO_2_, as the oxidation continued further [44].

It is important to note that in the present study, the Zy-4 alloy iwa heat-treated under 100 MPa at 310 °C, in contrast to the above studies, which can influence the oxidation behavior of Zr-based alloys. This can be explained, on one hand, by the insufficient amount of oxygen to promote zirconium oxidation. Thus, although the oxygen content detected on the surface was approximately 40% for both samples, it also oxidized the large amount of carbon found at the surface (Table 5). Furthermore, after 3024 h, the metallic feature decreased with increasing Zr^3+^, but Zr^2+^ had rather constant behavior (Table 4). Therefore, the oxygen was involved in the additional oxidation of zirconium, removing the metallic state completely.

On the other hand, the oxygen signal came from the outermost surface layer (<10 nm) due to the XPS surface sensitivity. Changes in oxygen chemistry are illustrated in Figure 11 and Figure 12.

Thereby, from the quantitative perspective, the oxygen showed a similar concentration in both samples of about 40% (Table 5), but its chemical responses on the surface were different—displaying the increasing tendency of lattice oxygen (O^2−^) with increasing oxidation time (Figure 11).

After separating oxygen’s chemical states (Figure 12, Table 6), three contributions were visible: adsorbed oxygen on the surface (Oads), oxygen bonded into the lattice as oxides (O^2−^) and OH groups [40,45]. The large amounts of OH groups accumulated on the surfaces of both samples were probably associated with the LiOH solution exposure. A clear difference lay in the adsorbed oxygen depletion, from 13.2% to 2.2%, accompanied by an increased oxide component, because of heat-treatment.

Therefore, high-pressure and high-temperature conditions could promote dissolution of oxygen into the bulk, or the conversion of adsorbed oxygen into surface lattice oxygen [41,43]. Moreover, referring to the study of Kaufmann [42], decomposition of ZrO_2_ to form suboxides or ZrO_2_ dissolution into the bulk could be favored by extreme conditions of temperature and pressure, as in the autoclaving process.

### 3.3. Electrochemichal Characterization

#### 3.3.1. Electrochemical Impedance Spectroscopy

For electrochemical evaluation of the protective properties of the oxides developed on the Zy-4 alloy surfaces, electrochemical impedance spectroscopy was used. The spectra recorded at open circuit potential after 10 min of immersion in LiOH solution are presented as Nyquist and Bode diagrams in Figure 13. As we can see from the Nyquist diagrams (Figure 13a), a single open capacitive appeared for all four types of sample we studied. For all anodized samples, higher values of the capacitive semicircle diameter were recorded compared to the non-anodized alloy.

From the Bode plots, Figure 13b, a higher impedance magnitude for the oxidized samples than for the unoxidized sample can be seen. The impedance magnitudes of the oxides developed after the oxidizing periods have a very high correlation—slight divergence is noticeable in the range of high frequencies (>10^4^).

As is known, impedance, |Z|, is directly proportional to oxide resistance and inversely proportional to oxide capacity, so a high impedance value indicates good corrosion resistance. In Figure 13, we can see the impedance magnitude of 5 × 10^6^ for the oxidized samples and 5 × 10^4^ for the unoxidized sample. Accordingly, the oxide layers developed on the surfaces of the tested samples offer better corrosion resistance.

EIS plots are considered to fit the equivalent circuit shown in Figure 14.

In this equivalent electrical circuit, Rs—solution resistance between the electrode and the electrolyte; CPEdl—constant phase element for double layer; Rct—charge transfer resistance; CPEoxt—constant phase element; Rox—resistance of the oxide layer. It is noteworthy that constant phase elements (CPE) must be introduced to explain the deviation of the capacitances between the actual measurements and the ideal pure capacitances, due to the local inhomogeneities of the dielectric material, surface roughness and relaxation effect; and the degree of deviation from the ideal capacitance depends on the value of *n*(0 < *n* < 1). A good fit of the data to this model was obtained, and the fitting parameters are listed in Table 7.

As we can see from Table 7, Rox increased with test duration. This means that the oxide film growth is stable, so the oxide film protects the alloy against corrosion. The thicker the layer, the lower the corrosion rate. The two constant phase elements have an almost capacitive behavior, because the values recorded for CPE-P were close to 1.

Platt et al. [46] reported that there was compressive stress in the oxide film during the zirconium oxide growth, and the stress increased as the oxide layer becomes thickened. Investigations into the behavior of zirconium alloy following exposure to LiOH have shown the presence of a non-protective porous film resulting from internal stress relaxation [47,48].

It was observed that the highest value for Rox was obtained for the thermally oxidized Zy-4 sample for 1512 h. This proves that the densest oxide layer was obtained then, which protected the alloy against corrosion.

#### 3.3.2. Potentiodynamic Polarization Tests

The potentiodynamic polarization curves of the oxidized samples are illustrated in Figure 15.

The corresponding electrochemical parameters deduced from the Tafel extrapolation polarization curves are listed in Table 8. The main parameters are the corrosion potential (E_corr_), corrosion rate (V_corr_), current density (i_corr_) and polarization resistance (Rp).

The protection efficiency (P_i_) and oxide porosity (P) have been evaluated quantitatively using Equations (3) and (4) respectively:(3)$$P_{i}=[1-(i_{cor}/i_{cor}^{o})]\cdot 100$$
(4)P=(RpsRp)·10−(ΔEcorβa)

As can be seen from Figure 15 and Table 8, for all thermally oxidized Zy-4 samples, lower values of corrosion current density were obtained compared to the unoxidized sample. Increasing the oxidation time led to a decrease in the corrosion rate, and the polarization resistance increased as the oxidizing time increased. The increase in the value of the corrosion density for the heat-treated sample for 3024 h was due to the cracks that appeared in the oxide layer, cracks that could be observed in the SEM analysis.

The oxide porosity and the protection efficiency are two important parameters for evaluating the oxide film’s integrity. The Pi and P results are also in agreement with the other results.

As shown in Figure 5, in the case of the sample oxidized for 3024 h, the oxide layer was characterized by large numbers of short lateral cracks and penetrating radial cracks. This mechanical degradation was responsible for the value from Table 8, and this behavior is comparable to the low temperature post-transition behavior of Zircaloy-4 cladding due to the “breakaway effect” under reactor operation conditions, as was observed in several studies about the mechanism of degradation during Zircaloy-4’s long-term exposure with undulated oxide growth, explaining the Pi increases and the P decreases [49,50,51].

It can also be seen that the best protection against corrosion was provided by the sample that was thermally oxidized for 1512 h. The highest value of porosity for the oxide layer being obtained for the 3024 h sample reconfirms the high value of the corrosion current density obtained for this sample.

#### 3.3.3. Open Circuit Potential Measurements

The open circuit potential (OCP) variation gives qualitative information about the oxide layer’s behavior, and it can predict the generalized corrosion behavior of samples. In Figure 16 we see that that the initial OCP for the unoxidized Zy-4 sample had a more cathodic value (−530 mV). For the heat-treated samples, stabilized OCP values were obtained: −100 mV for the sample oxidized for 504 h, and −50 mV for the Zy-4 sample exposed to LiOH for 3024 h. Additionally, no variations existed in the potential evolution. A slight increase can be observed. In agreement with these values, we can say that the oxide layer improves the corrosion behavior.

The results are in agreement with the literature [49,50,51,52,53], according to which after film breakdown during long-term exposure, undulated film growth takes place, not a direct transition to another kinetic law.

## 4. Conclusions

The morphological and electrochemical behaviors of oxide formed on Zy-4 alloy under simulated PHWR primary water conditions (310 °C, 10 MPa) were studied.

Gravimetric analysis showed that the oxidation of Zy-4 alloy under simulated PHWR primary water conditions follows a cubic law. As the oxidation period increased, the thickness of the oxides increased, and the corrosion rates decreased; and the corrosion rates decreased and increased again as an expression of the break down and film reformation. After this transition process, a post-transition with a reduced corrosion rate was observed. In fact, this process repeats itself in a cycle.

The presence of small hydrides uniformly distributed in the horizontal orientation across the Zy-4 alloy was highlighted.

The thickness of the oxide layer formed on the zircalloy surface, including changes in different regions, was estimated through gravimetric studies, but it was also measured by performing SEM on cross-sections of the samples, yielding quite similar values.

The microhardness tests, from oxide to alloy, indicated a slight decrease of Vickers microhardness values as a function of oxidation time.

The modifications of the morphology on the oxidized surfaces were highlighted. The autoclaving process combined with the LiOH exposure of Zy-4 alloys induced significant changes in surface area compared to the unoxidized sample. The hydrides were dispersed relatively homogeneously across each surface.

XPS measurements revealed the presence of all five oxidation states of zirconium: Zr^0^ Zr^+^, Zr^2+^ and Zr^3+^ and fully oxidized zirconium, Zr^4+^. Additionally, the increase of the heat-treatment duration from 504 to 3024 h led to a difference in the chemical behavior of zirconium, quantitatively, leading to an increasing Zr^3+^ contribution (Zr_2_O_3_), from 31.4% to 54.5%, accompanied by a decrease of metallic zirconium, from 9.3% to 3.0%. At the same time, Zr^4+^ (ZrO_2_) vanished. The oxygen concentration on the surface remained constant during the two stages of the thermal process.

The large number of OH groups accumulated on the surfaces of both samples was probably associated with the LiOH solution exposure. Therefore, high-pressure and high-temperature conditions could promote dissolution of oxygen into the bulk, or the conversion of adsorbed oxygen into surface lattice oxygen.

EIS impedance spectra show increases in the oxide impedance for the samples oxidized for longer periods of time. EIS plots had the best fit with an equivalent circuit, which illustrated an oxide model with two interfaces.

The best performances in terms of corrosion resistance were obtained for the sample that was thermally oxidized for 1512 h. When increasing the porosity of the oxide layer, in the case of the sample that was oxidized for 3024 h, it led to slightly higher values for corrosion current density, which could be explained based on undulated oxide growth after breakdown of film as an expression of passivation and depassivation processes. For all the oxidized samples, lower corrosion rates were obtained compared to the unoxidized sample.

In order to be able to confirm the evolution in time of the Zy-4 samples, several tests with longer oxidation times, along with the repetition of some of the samples subjected to thermal oxidation in this work, will be done in the near future.

## Figures and Tables

**Figure 1 materials-14-04586-f001:**
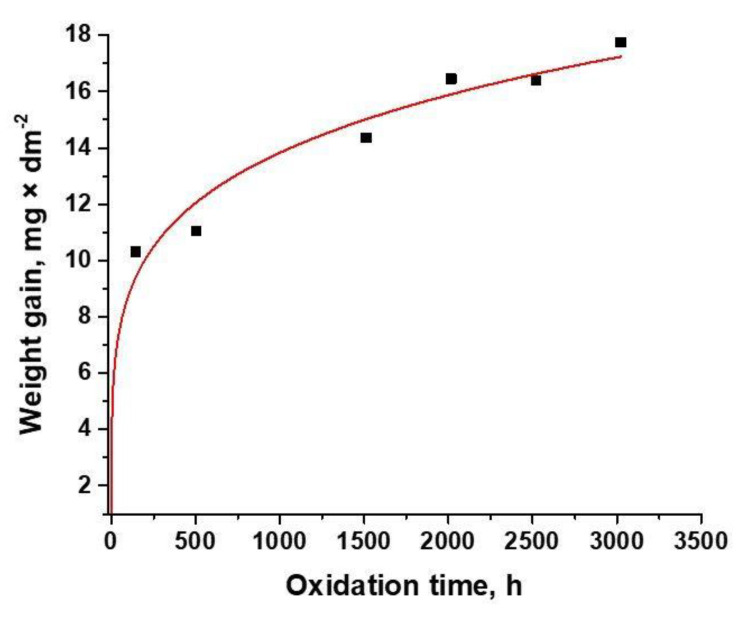
Oxidation kinetics of Zy-4 samples in the LiOH solution at 310 °C and 10 MPa.

**Figure 2 materials-14-04586-f002:**
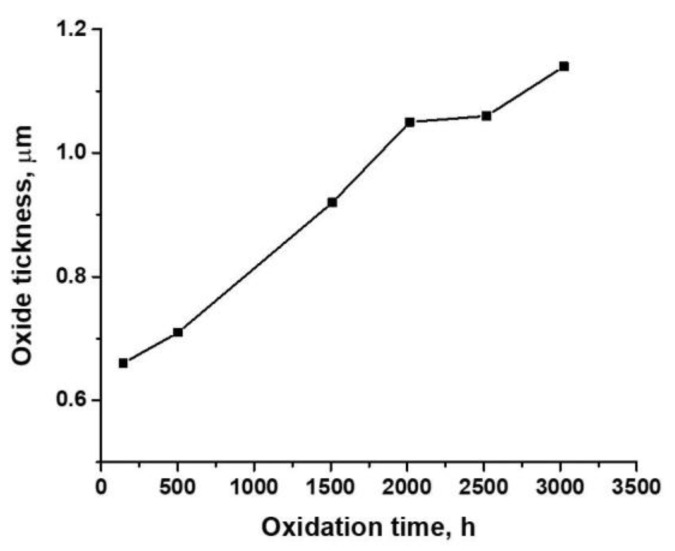
Oxide thickness’s dependency on exposure time for Zy-4 samples in a LiOH solution.

**Figure 3 materials-14-04586-f003:**
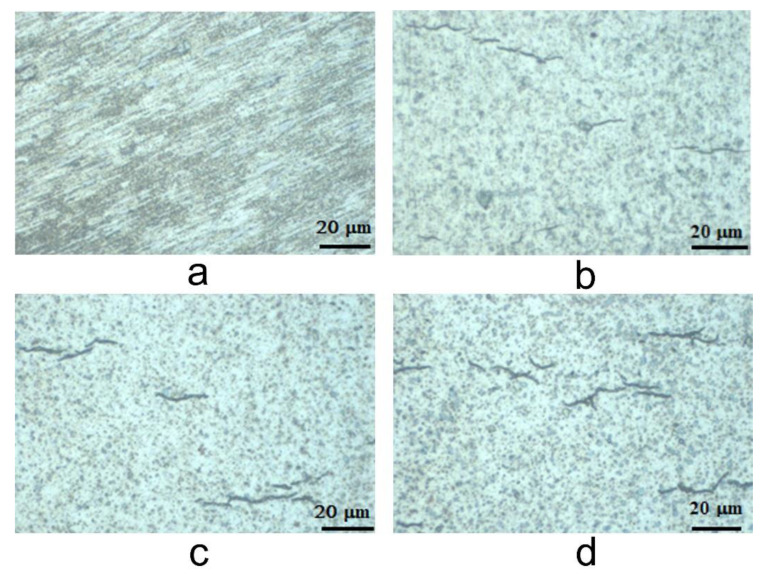
Representative Zy-4 hydride formation lengths of exposure to the LiOH solution: (**a**) 0 h; (**b**) 504 h; (**c**) 1512 h; (**d**) 3024 h.

**Figure 4 materials-14-04586-f004:**
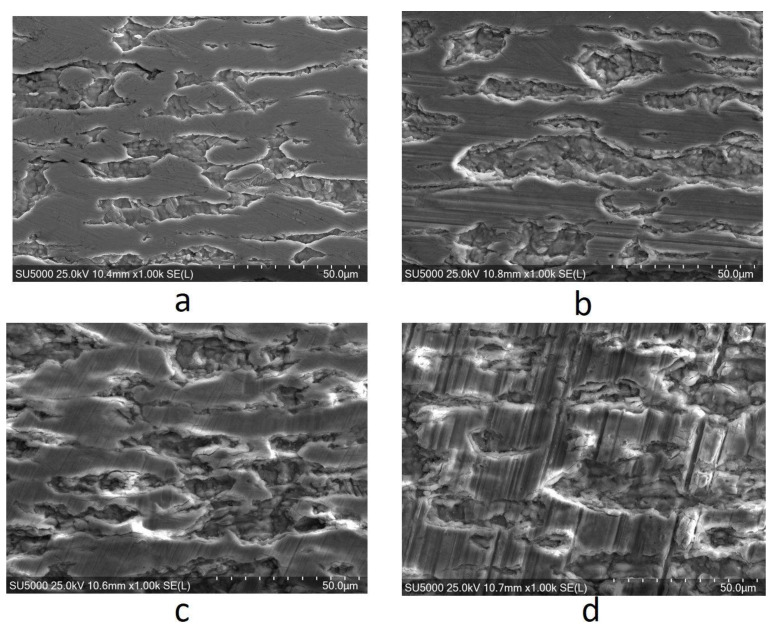
The surface morphologies of Zy-4 samples after different lengths of exposure to the LiOH solution: (**a**) 0 h, (**b**) 504 h, (**c**) 1512 h, (**d**) 3024 h.

**Figure 5 materials-14-04586-f005:**
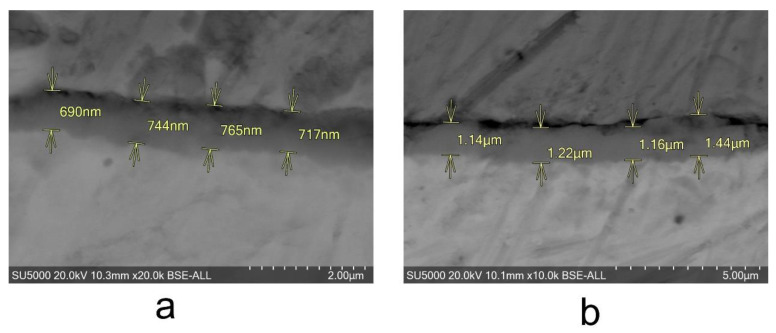
The surface morphologies for Zy-4 samples after different lengths of exposure to the LiOH solution: (**a**) 504 h and (**b**) 3024 h.

**Figure 6 materials-14-04586-f006:**
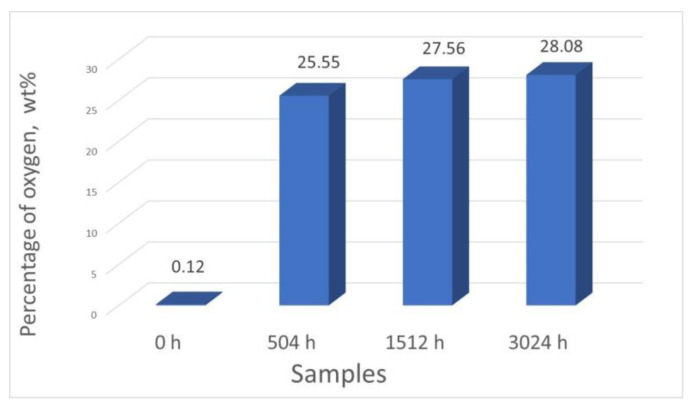
Comparison between the oxygen percentages of all the analyzed samples.

**Figure 7 materials-14-04586-f007:**
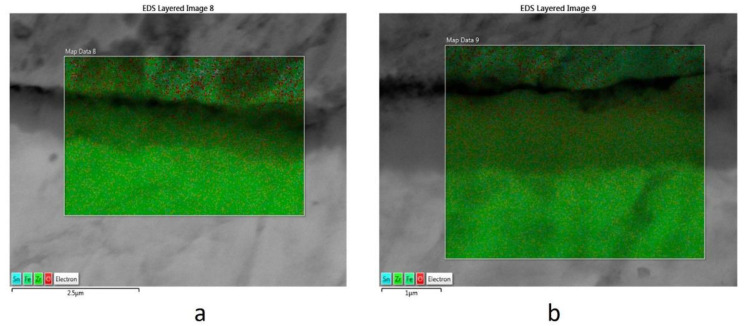
The surface mapping for Zy-4 samples after different lengths of exposure to the LiOH solution: (**a**) 504 h, (**b**) 3024 h.

**Figure 8 materials-14-04586-f008:**
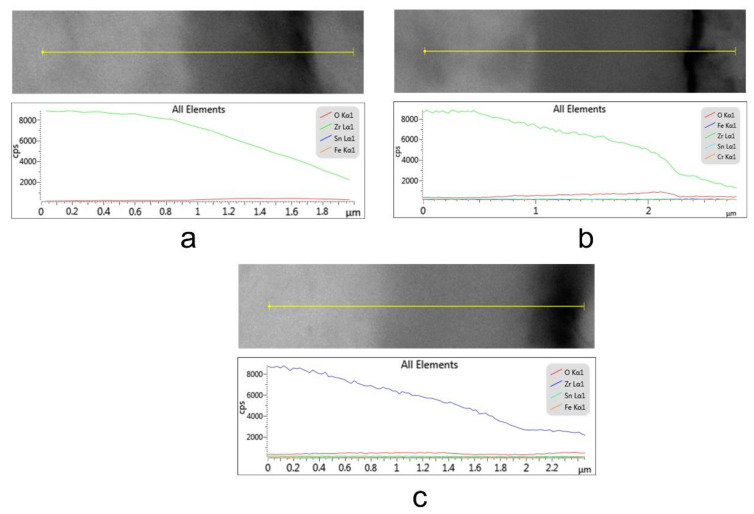
EDS profile of a cross-section through an oxide layer grown on an Zy-4 sample after different lengths of exposure to the LiOH solution: (**a**) 504 h, (**b**) 1512 h, (**c**) 3024 h.

**Figure 9 materials-14-04586-f009:**
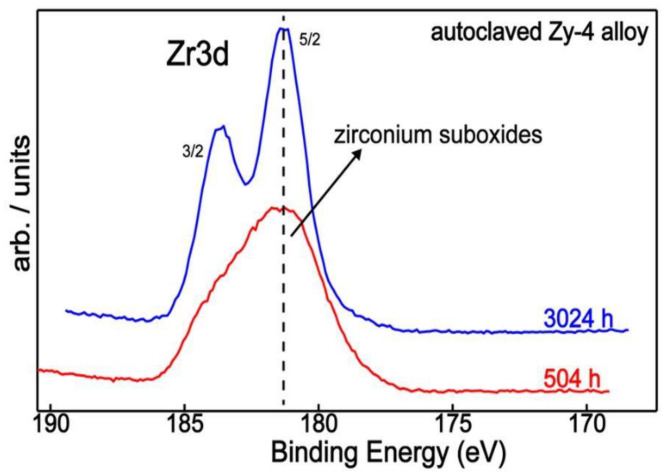
Zr3d XPS superimposed spectra of the Zy-4 alloy after 504 or 3024 h of autoclaving.

**Figure 10 materials-14-04586-f010:**
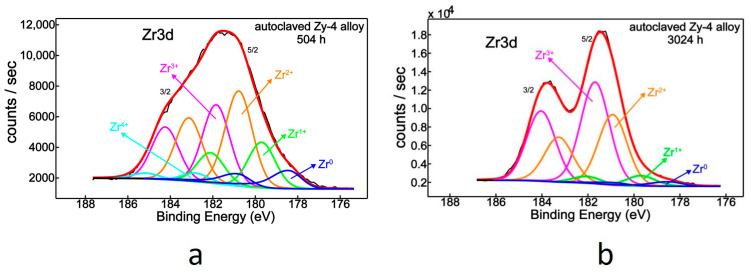
Zr3d XPS deconvoluted spectrum for Zy-4 alloy after (**a**) 504 h or (**b**) 3024 h of autoclaving.

**Figure 11 materials-14-04586-f011:**
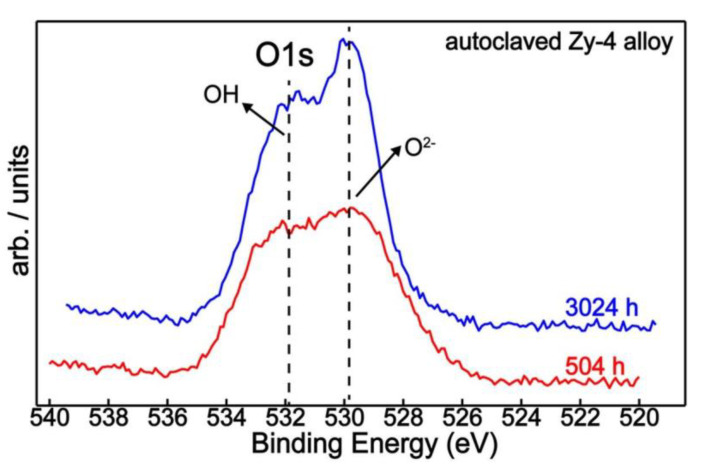
O1s XPS superimposed spectra of Zy-4 alloy after 504 and 3024 h of autoclaving.

**Figure 12 materials-14-04586-f012:**
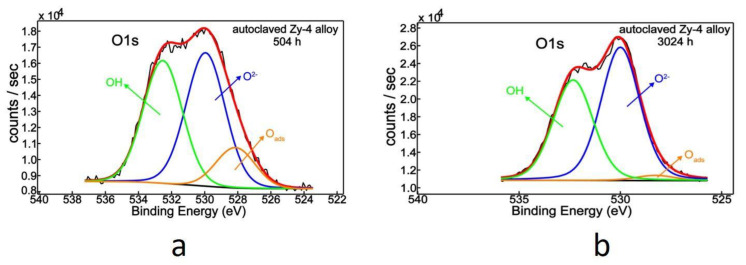
O1s XPS deconvoluted spectra for Zy-4 alloy after 504 h (**a**) and 3024 h (**b**) of autoclaving.

**Figure 13 materials-14-04586-f013:**
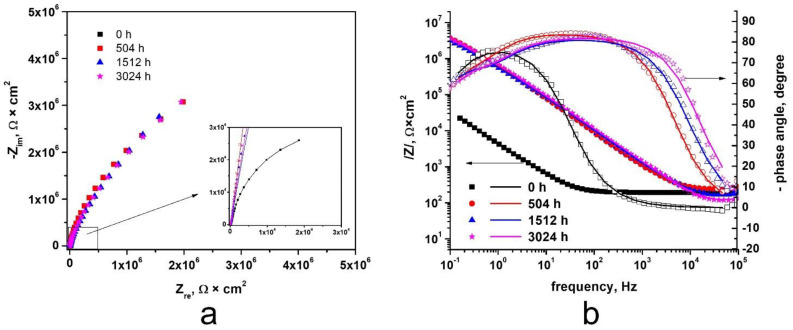
Nyquist (**a**) and Bode (**b**) diagrams for Zy-4 samples after different oxidation times in LiOH solution at 310 °C and 10 MPa. Symbols show experimental data, whereas lines represent fitted data using the electrical equivalent circuit.

**Figure 14 materials-14-04586-f014:**
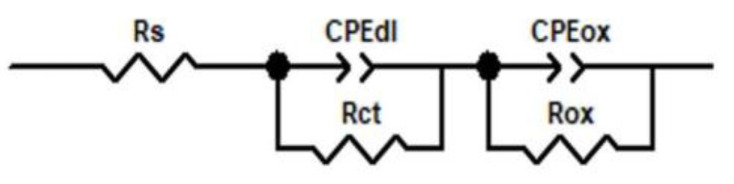
Equivalent circuit diagram for the oxide film formed on Zircaloy-4 in the LiOH solution after different oxidizing times at 310 °C and 10 MPa.

**Figure 15 materials-14-04586-f015:**
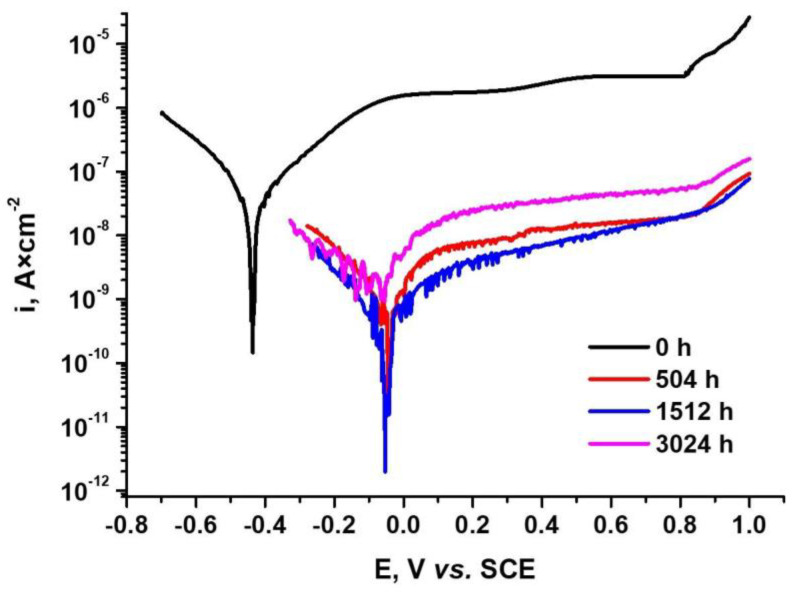
Comparison of polarization curves for unoxidized Zy-4 alloy and alloy oxidized for 504, 1512 or 3024 h in our LiOH solution (310 °C and 10 MPa) at a scan rate of 0.5 mV·s^−1^.

**Figure 16 materials-14-04586-f016:**
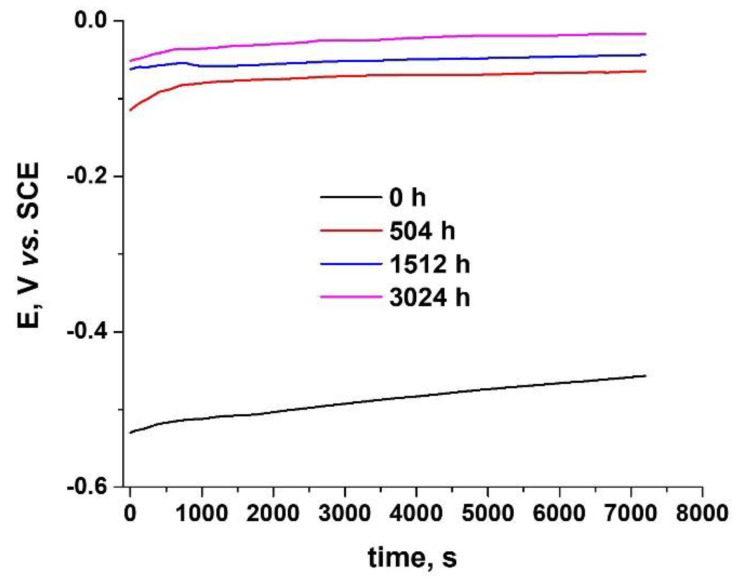
Variation over time of OCP for unoxidized Zy-4 alloy and alloy oxidized for 504, 1512 or 3024 h in our LiOH solution at 310 °C and 10 MPa.

**Table 1 materials-14-04586-t001:** Composition of Zircaloy tube alloy (wt.%).

Alloying Elements, [wt.%]				
Sn	Fe	Cr	O	Zr
1.32	0.29	0.14	0.12	Balance

**Table 2 materials-14-04586-t002:** Kinetic parameters for Zy-4 samples.

Kinetic Equation	k_p_	*n*	R^2^
y = 3.49 × t ^0.199^	3.49	0.199	0.987

**Table 3 materials-14-04586-t003:** Composition expressed in weight percent (wt.%) (elemental analysis).

Sample Oxidized for	Zr	O	Sn	Cr	Fe
504 h	73.04	25.55	0.99	0.13	0.29
1512 h	71.12	27.57	0.94	0.10	0.27
3024 h	70.64	28.08	0.88	0.13	0.26

**Table 4 materials-14-04586-t004:** Surface chemistry of zirconium: chemical states and their relative concentrations.

Sample	Binding Energy (eV)	Zirconium Chemical Species	Zirconium Relative Concentrations (%)
Zy-4 (504 h)	178.5 179.7 180.8 181.8 182.8	Zr^0^ Zr_2_O ZrO Zr_2_O_3_ ZrO_2_	9.3 18.1 37.8 31.4 3.4
Zy-4 (3024 h)	178.5 179.7 180.9 181.7	Zr^0^ Zr_2_O ZrO Zr_2_O_3_	3.0 5.8 36.7 54.5

**Table 5 materials-14-04586-t005:** Element relative concentrations (at.%).

Sample	C1s	O1s	Zr3d
Zy-4 (504 h)	50.7	40.5	8.8
Zy-4 (3024 h)	50.7	40.7	8.6

**Table 6 materials-14-04586-t006:** The surface chemistry of oxygen: chemical states and their relative concentrations.

Sample	Binding Energy (eV)	Oxygen Chemical Species	Oxygen Relative Concentrations (%)
Zy-4 (504 h)	528.2 530.0 532.5	O_ads_ O^2−^ OH	13.2 45.6 41.2
Zy-4 (3024 h)	528.3 530.0 532.3	O_ads_ O^2−^ OH	2.2 55.5 42.3

**Table 7 materials-14-04586-t007:** The values of equivalent electrical circuits elements for unoxidized and oxidized Zy-4 samples, which spent different lengths of time in our LiOH solution at 310 °C and 10 MPa.

Sample after Different Oxidation Time, h	R_s_, Ω·cm^2^	CPE_dl_-T μF·cm^−2^	CPE_dl_-P	R_ct_ Ω·cm^2^	CPE_ox_-T μF·cm^−2^	CPE_ox_-P	R_ox_ KΩ·cm^2^	Chi-Squared
0	193	42.9	0.89	83,482	-	-	-	4.4 × 10^−3^
504	232.8	0.347	0.92	8.72 × 10^6^	1.07	0.99	174	2.1 × 10^−3^
1512	155.5	0.411	0.89	9.19 × 10^6^	0.87	0.95	176	3.5 × 10^−3^
3024	103.6	0.252	0.93	9.39 × 10^6^	0.96	0.93	40.42	5.9 × 10^−3^

**Table 8 materials-14-04586-t008:** Polarization parameters of Zircaloy-4 oxidized for samples oxidized in LiOH solution (310 °C, 10 MPa).

Sample after Different Oxidation Time, h	E_corr_, mV	i_corr_, µA·cm^−2^	V_corr_ Mm·year^−1^	R_p_ MΩ·cm^2^	P_i_ (%)	P (%)
0	−436	1.9 × 10^−1^	6.17 × 10^−4^	0.26	^-^	-
504	−63.4	1.62 × 10^−2^	1.07 × 10^−5^	5.9	91.47	0.14
1512	−90.8	4.65 × 10^−3^	3.71 × 10^−5^	19	97.55	0.05
3024	−67.8	4.25 × 10^−2^	9.22 × 10^−4^	4.2	77.63	1.5

## Data Availability

The data presented in this study are available upon request.

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
