# Peer review of "Long-Term Corrosion Testing of Zy-4 in a LiOH Solution under High Pressure and Temperature Conditions"

_materials, 2021, doi:10.3390/ma14164586_

Round 1
Reviewer 1 Report
Manuscript ID: materials-1312515
Title: Long-Term Corrosion Testing of Zy-4 in a LiOH Solution at High Pressure and Temperature Conditions
Authors: Diana Diniasi , Florentina Golgovici * , Alexandru Marin , Aurelian Denis Negrea , Manuela Fulger , Ioana Demetrescu
Review Report
Line 87: I assume that Zy-4 is an abbreviated designation for the material Zircaloy-4, state it there in the line 87 or 90.
Line 134-138: Correct the sentence.
Figure 3: Pictures e, f and g, h are missing !
Table 8: Correct the table, column Vcorr , 6.17 . 10-4 or 6.17 x 10-4 ? Unify multiplication sign.
Lines 439-448: You write.....So we can see that as the oxidizing time increase, the Pi increases and the P decreases. But this does not apply in the case of 3024 h. In general, the results for 3024 h do not confirm the rules you are writing about.
I would expect an increase in Ecor, a lower value of Vcorr, a higher Rp. And also a higher value of Pi and, conversely, a lower value of P.
Explain, why is there an opposite trend for the oxidation time of 3024 h?
Or it is necessary to take a new measurement.
Fig. 16: On the OCP graph, we see a shift to more positive potentials with oxidation time. It does not correlate with the data in Table 8.
Lines 492-498: Please, explain more clearly. Take into account my comments mentioned above.
Reviewer 2 Report
Dear Authors,
Overall, this is a clear and well-written manuscript. The introduction is relevant and results are clear and well explained. There are number of comments and suggestion to improve the manuscript. This manuscript could be accepted after minor revision.
Line 97, 362,… what is the concentration of LiOH in solution? It mentioned many times in text without given concentration.
135-138, should rewrite this sentence it is not clear.
183, thickness dependency
212, image of g and h is missing. Scale bar in the image should be the same size and type.
225, figure 4d, why the hydride cracking show the morphology like polished surface? Author claimed the strip-lines occurred due to exposed surface to LiOH but in Fig. 3a we could see similar strip lines. In my opinion, this explanation is not correct about the changing morphology due to generation of hydrides. What is clear the thickness of formed layer increased without change in morphology….
239, the color that represent the thickness in figure is not readable please change it to bright color like yellow or white
255-257, need to rewrite.
265, the graphs are not clear, hardly could read the scales. Make better graphs with better resolution.
402-406, need to rewrite.
415, scan rate is missing from fig caption
447, author introduce new term of anodized sample… which is wrong. We are not dealing with anodization process
453 oxidized sample (-50 …-100mV)????
524, 533, 584 and more… there is no consistency in references. Should use the same format.
Reviewer 3 Report
In general I find the article interesting. I suggest making a few minor changes. There are comments in the text.

Reviewer 4 Report
This study is of importance but it is more suitable for the Journal Metals. So please submit it to Metals.
In addition, the conclusion should be concise and In Figure 9 and Figure 11, the XPS peak may be divided.
Round 2
Reviewer 1 Report
Manuscript ID: materials-1312515
Title: Long-Term Corrosion Testing of Zy-4 in a LiOH Solution at High Pressure and Temperature Conditions
Authors: Diana Diniasi , Florentina Golgovici * , Alexandru Marin , Aurelian Denis Negrea , Manuela Fulger , Ioana Demetrescu
Review Report 2
Tables 7, 8: Check the tables and unify the characters, for example mA*cm-2, mF*cm-2, Ω·cm2
unify the characters * , . , x
Lines 458-469: You tried to explain the opposite trend for an oxidation time of 3024 h. That's true, but in my opinion it is not appropriate to compare the results for 3024 h with the previous ones.
OK, this is a sufficient explanation for this post, but I would recommended to repeat the measurement in the future.
Please, add to the conclusion, it would be appropriate in the future to focus on repeating some measurements in order to confirm your explanations.
Reviewer 4 Report
The quality of this paper should be further improved.
